# Needle-Free Jet Injectors and Nanosuspensions: Exploring the Potential of an Unexpected Pair

**DOI:** 10.3390/pharmaceutics14051085

**Published:** 2022-05-19

**Authors:** Michele Schlich, Luca Casula, Aurora Musa, Rosa Pireddu, Giulia Pitzanti, Maria Cristina Cardia, Donatella Valenti, Salvatore Marceddu, Anna Maria Fadda, Maria Antonietta De Luca, Chiara Sinico, Francesco Lai

**Affiliations:** 1Dipartimento di Scienze della Vita e dell’Ambiente, Sezione di Scienze del Farmaco, CNBS, Università degli Studi di Cagliari, 09124 Cagliari, Italy; michele.schlich@unica.it (M.S.); luca.casula@unica.it (L.C.); rosapireddu@unica.it (R.P.); cardiamr@unica.it (M.C.C.); valenti@unica.it (D.V.); sinico@unica.it (C.S.); frlai@unica.it (F.L.); 2Department of Biomedical Sciences, University of Cagliari, 09042 Monserrato, Italy; au.musa16@gmail.com (A.M.); deluca@unica.it (M.A.D.L.); 3School of Pharmacy, Queen’s University Belfast, Belfast BT9 7BL, UK; giuliapitzanti@tiscali.it; 4Istituto di Scienze delle Produzioni Alimentari (ISPA)-CNR, Sez. di Sassari, 07040 Baldinca, Italy; salvatore.marceddu@cnr.it

**Keywords:** nanocrystals, needle-free, subcutaneous, medical device, diclofenac

## Abstract

Needle-free liquid jet injectors are medical devices used to administer pharmaceutical solutions through the skin. Jet injectors generate a high-speed stream of liquid medication that can puncture the skin and deliver the drug to the underlying tissues. In this work, we investigated the feasibility of using liquid jet injectors to administer nanosuspensions, assessing the impact of the jet injection on their pharmaceutical and physicochemical properties. For this purpose, the model drug diclofenac was used to prepare a set of nanosuspensions, stabilized by poloxamer 188, and equilibrated at different pHs. The hydrodynamic diameter and morphology of the nanocrystals were analyzed before and after the jet injection across porcine skin in vitro, together with the solubility and release kinetics of diclofenac in a simulated subcutaneous environment. The efficacy of the jet injection (i.e., the amount of drug delivered across the skin) was evaluated for the nanosuspension and for a solution, which was used as a control. Finally, the nanosuspension was administered to rats by jet injector, and the plasma profile of diclofenac was evaluated and compared to the one obtained by jet injecting a solution with an equal concentration. The nanosuspension features were maintained after the jet injection in vitro, suggesting that no structural changes occur upon high-speed impact with the skin. Accordingly, in vivo studies demonstrated the feasibility of jet injecting a nanosuspension, reaching relevant plasma concentration of the drug. Overall, needle-free jet injectors proved to be a suitable alternative to conventional syringes for the administration of nanosuspensions.

## 1. Introduction

Needle-free jet injectors (NFJIs) are effective alternatives to conventional syringes for the administration of liquid medications across the skin [1]. Their functioning principle relies on the application of pressure in a confined chamber containing the pharmaceutical solution. A micro-nozzle on the bottom of the chamber held against the skin allows the exit of the liquid as a high-speed jet which is able to puncture the skin and reach the dermis, the subcutaneous, or the muscular region, depending on the injector parameters [2]. The substitution of conventional syringes with NFJIs comes with advantages in terms of reduced production of hazardous wastes, improved operator safety, and patient acceptability. Specifically, the disposable part of NFJIs is a small plastic ampoule, while each conventional injection produces a sharp needle to be discarded with care. The risk of needle stick injuries, common among healthcare practitioners during the opening and disposal of a conventional syringe, is eliminated when using an NFJI [3]. Lastly, NFJIs allow patients to overcome their fear of needles (defined as needle-phobia), a condition that may lead the patients to delay and discontinue injection-based treatments, with possibly severe consequences on their health conditions [4]. For all of these reasons, NFJIs are gaining popularity, and are clinically employed to administer vaccines and drugs that require systemic (hormones, antibiotics, anti-migraine) or local (anesthetics, anti-inflammatory agents) action [5,6,7,8,9]. Importantly, NFJIs are playing a role in the global vaccination campaign against SARS-CoV-2. The NFJI Tropis (PharmaJet^®^) will be the exclusive administration method of a plasmid DNA vaccine, which was recently approved in India [10]. Additionally, the use of Comfort-in™ (Mika Medical Co.), the same device we used in this study, was pioneered in Europe by an Italian vaccination hub, following its successful use in the USA, Australia, and India [11]. Most of the drugs currently administered through NFJIs are formulated as clear solutions, with an exception represented by triamcinolone acetonide, available as a suspension [12]. However, a large number of innovative injectable formulations based on nano- and microparticles dispersed in a liquid vehicle are under development, with many examples already authorized by regulatory agencies for clinical use [13]. For these nano-/micro-drug delivery systems, it is mandatory to ensure that the size, shape, and surface properties of the particles are maintained during the administration procedure, since such properties dictate the drug release, tissue distribution, and cell uptake. In the case of NFJIs, the pressure built into the chamber and the high-speed collision with the skin might have an impact on the physicochemical and pharmaceutical properties of nano- and microparticles, raising concerns on the possible use of NFJIs for the administration of these formulations. Indeed, the combination of NFJIs and nanoparticles has only been explored in a limited number of studies, with the main results summarized in Table 1 [14,15,16,17,18,19]. Previously, we determined that flexible and conventional liposomes administered across the skin by an NFJI maintain their structural and functional properties [17]. In this work, we focus our attention on the possible combination of NFJI with another nano-delivery system: nanosuspensions. Nanosuspensions are defined as dispersions of nanometric particles of a pure drug in a liquid vehicle [20]. Drug nanocrystals can be prepared by top-down or bottom-up methods, and are usually stabilized by one or more surfactants or polymers [21,22]. Despite the majority of marketed nanocrystals being approved for oral administration due to their efficient enhancement of bioavailability [23,24], the parenteral and topical routes hold great promise for the treatment of loco-regional pathologies, such as joint and muscle inflammation. In our previous works, we demonstrated the transdermal delivery of anti-inflammatory drugs formulated as nanosuspensions, either alone or in combination with medical devices (i.e., microneedle rollers) [25,26]. With the aim of expanding the delivery methods of nanosuspensions, here, we demonstrate the feasibility of using an NFJI (Comfort-in™) to administer a diclofenac nanosuspension (DCF NS) across the skin, without affecting the formulation properties. 

## 2. Materials and Methods

### 2.1. Materials

Diclofenac sodium salt and sodium hyaluronate (low molecular weight) were purchased from Galeno (Comeana, Italy). Poloxamer 188 (P188). All other reagents and solvents were purchased from Sigma–Aldrich (Milan, Italy), and used without further purification. A commercial jet injector, Comfort-in™, equipped with disposable 0.5 mL nozzles was generously gifted by Gamastech S.R.L. (Sant’Agata li Battiati, Italy).

### 2.2. Synthesis of Diclofenac Acid and Preparation of Nanosuspensions

The acid form of diclofenac was prepared according to a published procedure which allows for the production og a crystalline solid [27]. Briefly, diluted hydrochloric acid was added to a saturated water solution of diclofenac sodium salt to form a white precipitate of diclofenac acid. The addition of HCl was continued in portions until no additional precipitate was formed. The solid was filtered, washed thoroughly with distilled water, and air dried. The DCF nanosuspensions were prepared according to the schematic in Figure 1A, using a high-speed homogenization/wet media milling combined technique. The bulk drug was weighted and dispersed in an aqueous solution of P188 (5 mg/mL), using an Ultra Turrax T25 basic for 5 min at 6500 rpm. The obtained suspension, with a concentration of 10 mg/mL DCF and 5 mg/mL P188, was transferred in 1.5 mL conical tubes containing approximately 0.4 g of 0.1–0.2 mm yttrium-stabilized zirconia–silica beads (Silibeads^®^ Typ ZY Sigmund Lindner, Warmensteinach, Germany). The tubes were oscillated at 3000 rpm for 60 min using a bead-milling cell disruptor device (Disruptor Genie^®^, Scientific Industries, Bohemia, NY, USA). The obtained nanosuspensions were separated from the milling beads by sieving. The pH of the formulations was measured by a pH-meter (Mettler Toledo FiveEasy™, Switzerland), and adjusted by discrete additions of NaOH (1.25 M) or citric acid (0.26 M) solutions. A diclofenac sodium solution (10 mg/mL) in deionized water was also prepared and used as a control when specified.

### 2.3. NFJI-Mediated Delivery across the Skin

A setup was designed to allow the jet injection of DCF nanosuspensions across the skin and its subsequent recovery for analysis. Full-thickness skin was obtained from newborn Goland–Pietrain hybrid pigs (1–1.5 kg) which had died of natural causes, provided by a local slaughterhouse. Upon receival of the pig cadaver, the skin was excised from the back side of the animal, thoroughly washed with saline, dried by blotting with tissue paper, and stored at −80 °C until use. Briefly, a circular pig skin specimen with a diameter of 2.5 cm was cut, pre-equilibrated with saline, and then placed on a disposable glass vial (10 mL volume), supported by a wire mesh, with the stratum corneum side on the top. The plastic ampoule of the jet injector was filled with 400 μL of a DCF nanosuspension, and the injection was performed holding the nozzle at a 90° angle to the skin, as recommended by the manufacturer of the NFJI. The jet-injected DCF nanosuspensions were recovered from the glass vial beneath the skin, and used for all the characterizations described in the following paragraphs.

### 2.4. Dynamic Light Scattering

The average diameter and polydispersity index (PDI) of nanosuspensions were determined by dynamic light scattering (DLS) using a Zetasizer Nano (Malvern Instruments, Worcestershire, UK). For DLS measurements, the DCF NS and the jet-injected DCF NS were diluted with deionized water (1:100) immediately before the analysis.

### 2.5. Scanning Electron Microscopy

The morphology of DCF NS before and after the jet injection was assessed by scanning electron microscopy (SEM). A 3 µL drop of DCF NS or jet-injected DCF NS was deposited on a silicon wafer and air dried. The sample was sputter coated with a 10 nm layer of gold, and imaged using a SEM (Jeol, Japan) operating at 10 kV. A Zeiss ESEM EVO LS 10 (Oberkochen, Germany) operating at variable pressure (VP) was employed to image the skin after the jet injection of DCF NS or of a diclofenac sodium solution. Skin samples were mounted onto aluminum stubs, and imaged without any pre-treatment, operating at 20 kV in VP.

### 2.6. Saturation Solubility

The DCF saturation solubility was measured for the DCF NS before and after the jet injection. The DCF NS or DCF NS recovered after the jet injection were kept under constant stirring for 48 h at 37 °C. Samples were withdrawn and centrifuged at 13,000 rpm for 60 min to precipitate the nanocrystals. The supernatant was centrifuged again at 13,000 rpm for 30 min. Then, 200 μL of the clear supernatant was withdrawn and diluted with 800 μL of methanol for the HPLC analysis.

### 2.7. Quantification of DCF Delivered

Either a DCF NS or a DCF sodium solution was jet-injected through the skin as described above. The efficacy of the jet injection was evaluated by quantifying the DCF recovered on the skin surface, within the skin, and inside the vial beneath the skin. The DCF on the skin surface was collected by blotting using three pieces of paper towel measuring 2.5 × 2.5 cm^2^. Briefly, a paper towel was carefully pressed against the skin surface, cut, and placed in a flask with methanol, and sonicated for 4 min to extract the drug. To ensure the complete adsorption of the drug, the procedure was repeated with three pieces of paper towel for each skin specimen. The obtained suspension was filtered out and assayed for drug content by HPLC. The skin specimen was cut and subjected to the same treatment to extract the DCF entrapped within the tissue. The DCF recovered from the vial (i.e., dose delivered across the skin) was dissolved in methanol, and diluted for HPLC analysis.

### 2.8. Release Studies in Simulated Subcutaneous Environment

A gel simulating the subcutaneous (SC) environment was prepared according to a published protocol [28]. Briefly, a buffer containing 6.4 mg/mL NaCl, 0.04 mg/mL MgCl_2_, 0.4 mg/mL KCl, 0.2 mg/mL CaCl_2_, and 2.1 mg/mL NaHCO_3_ (release buffer) was prepared, and its pH was adjusted to 7.4. The solution was then used to dissolve sodium hyaluronate (26.7 mg/mL), as well as to obtain a viscous gel (SC gel). Amounts of DCF NS or DCF NS recovered after the jet injection were mixed with the SC gel (1:3 volumes) by pipetting, followed by being inserted in a dialysis bag and incubated at 37 °C in 100 mL of the release buffer. At regular intervals, up to 24 h, 10 mL of release buffer were withdrawn, replaced with fresh buffer to ensure sink conditions, and analyzed by HPLC for drug content.

### 2.9. Pharmacokinetic Studies

Adult male Sprague–Dawley rats weighing 275–300 g (Envigo, Italy) were employed for in vivo pharmacokinetic experiments. Rats were housed in groups of six in standard conditions of temperature (21 ± 1 °C) and humidity (60%) under a 12 h/12 h light/dark cycle (lights on at 7.00 am), with food and water available ad libitum. All experiments were carried out in accordance with European Council directives (609/86 and 63/2010), and in compliance with the animal policies approved by the Italian Ministry of Health and the Ethical Committee for Animal Experiments (CESA, University of Cagliari, Italy). We made all efforts to minimize pain and suffering, and to reduce the number of animals used. Rats were anaesthetized with isoflurane gas (Merial, Milano, Italy), and were maintained under anesthesia using a breathing tube under a scavenging system for the implant of a polyethylene catheter in the right jugular vein, as previously described [29]. Animals were divided in two groups, and were treated with a single subcutaneous administration of either DCF sodium solution or DCF NS (6.65 mg DCF/kg) delivered by the Comfort-in™, following the manufacturer instructions. At different time points after the injection, blood samples (300 µL) were withdrawn through the venous catheter, and collected in heparinized tubes. The whole blood was centrifuged at 12,000 rpm for 15 min at room temperature, and the plasma was collected and transferred to new tubes. Drug extraction was obtained by treating the plasma samples with methanol, vortexing, and then centrifuging at 15,000 rpm for 15 min at 10 °C. The supernatant was collected and analyzed by HPLC, as described below.

### 2.10. Quantitative Determination of DCF

The quantification of DCF was performed by an Alliance 2690 (Waters Corp, Milford, MA) HPLC system equipped with a photodiode array detector and a computer software (Empower 3, Waters). DCF was determined at 280 nm using a SunFire C18 column (3.5 µm, 4.6 mm × 100 mm, Waters, Milan, Italy), and a mixture of 40.75% water, 59.225% acetonitrile, and 0.025% acetic acid (*v*/*v*) as a mobile phase, delivered at a flow rate of 0.5 mL/min. A calibration curve was built by using standard solutions (0.01–0.5 mg/mL) prepared by the dilution of a stock standard solution. The limit of detection was 1 ng, while the limit of quantification was 2 ng.

### 2.11. Statistical Analysis of Data

Results are expressed as the mean ± standard deviation. Student’s t-test was used to compare results from two samples. Data analysis was carried out with the software package XLStat for Excel (Microsoft, Redmond, WA, USA). Significance was tested at the 0.05 level of probability (p).

Data of in vivo studies are expressed as mean ± SEM, and differences were statistically significant at *p* < 0.05. Results were analyzed using repeated measures (RM) ANOVA, followed by Bonferroni’s post-hoc test. For RM tests, whenever we could not assume sphericity, a Geisser–Greenhouse correction was carried out. Statistical analysis of data from in vivo experiments was performed with GraphPad Prism 8 software (GraphPad Prism, San Diego, CA, USA).

## 3. Results and Discussion

The goal of this study was to assess the feasibility of using a commercial NFJI to administer pharmaceutical nanosuspensions. To this aim, various nanosuspensions of DCF were prepared, and their physicochemical and pharmaceutical properties were determined before and after the NFJI administration across a skin specimen. The DCF NS was prepared according to the established high-speed homogenization/wet media milling technique, followed by the adjustment of pH through the addition of citric acid or sodium hydroxide (Figure 1A). Modifying the pH of a weak acid nanosuspension (such as DCF) is a straightforward way to tune the solubility of the drug. Therefore, with the idea of providing a subcutaneous depot of drug nanocrystals, tuning the pH could provide higher or lower fractions of dissolved drug, influencing the rate of drug absorption. Freshly prepared DCF NS has a pH of 5.8 ± 0.1, and the adjustment to 4.0, 5.0, 6.0, and 7.0 did not show significant changes in the size and PDI of suspended nanocrystals (Figure 1B). Specifically, DCF NS (pH 6.0) has a size of 450 ± 67 nm which is retained in the range 435–473 upon pH adjustment. All four formulations were thus employed for the feasibility study with the NFJI.

The Comfort-in™ is a spring powered NFJI, clinically employed for the administration of solutions of local anesthetics, anti-inflammatory agents, vitamins, and insulin. Upon release of the trigger, the spring pushes a plunger against the liquid medication contained in a disposable ampoule, forcing it through a 200 µm nozzle at a pressure of 3900 psi. Such a “liquid needle” punctures the skin and delivers the drug to the subcutaneous region. The rationale behind the use of the Comfort-in™ with nanosuspensions lies in the combined advantages in terms of (1) patient acceptability and safety provided by the NFJIs; and (2) improvement of biopharmaceutical properties that can be achieved through rational design and preparation of NS. To ensure that no changes occur to the NS structural features that are intimately linked to their biopharmaceutical properties upon administration with the Comfort-in™, the mean diameter and morphology of the nanocrystals were determined after their jet injection across a skin specimen. Excised skin from newborn pigs was chosen for its similarity with human skin, and for its widespread use as a model for the assessment of transdermal permeation of drugs and nanoparticles [30]. In the present work, full-thickness porcine skin accurately modeled the texture and mechanical properties of human skin, providing a suitable model to assess the effect of the high-speed impact of jet-injected NS with the skin surface.

Figure 2 reports the mean diameter (Z average) and PDI of DCF nanocrystals before and after the jet injection across porcine skin. All of the DCF NS showed no significant differences in nanocrystals size upon the administration with the NFJI. As for the PDI, which is an indicator of the size distribution of nanocrystals in the sample, a trend towards higher values could be observed for the jet-injected samples compared to the non-injected DCF NS. This behavior suggests that the NFJI process might slightly broaden the size distribution of particles, resulting in a loss in homogeneity of the sample upon injection. However, it is important to note that such differences are not significant (*p* > 0.05) for all pairs of samples, except for the DCF NS pH 4.

To further investigate the effect of NFJI delivery on nanocrystal size distribution and morphology, the DCF NS was imaged by SEM before and after the jet injection across the skin (Figure 3). Dried DCF nanocrystals are a relatively homogenous dispersion of particles with a polyhedral shape, with a mean diameter in good agreement with the results of the DLS analysis. No evident changes can be detected in the shape and size distribution of DCF nanocrystals upon the injection of DCF NS across the skin (Figure 3C,D). A similarly low number of large particles can be observed in all of the samples (before and after the jet injection), leading us to conclude that they were not generated by aggregation upon injection, but rather were present in the initial sample due to sub-optimal milling. 

One of the key properties of nanosuspensions is their ability to increase the saturation solubility of hydrophobic drugs and, consequently, their bioavailability [31]. Comparing the saturation solubility of DCF from nanosuspensions before and after the jet injection provides additional information on the effect of jet injection on the nanocrystals’’ properties. Therefore, we included the analysis of saturation solubility at 37 °C in the panel of in vitro tests to assess the feasibility of using NFJIs for the administration of nanosuspensions. We measured the solubility of DCF in nanosuspensions adjusted to pH 4.0, 5.0, 6.0, and 7.0, before and after the jet injection (Figure 4). In general, the solubility of DCF increases with the pH due to the partial ionization of the drug, leading to a solubility range between 18 µg/mL (pH 4.00) and 739 µg/mL (pH 7.00). Most importantly for the aim of this work, the saturation solubility did not change following the jet injection of the DCF NS at any of the pH levels tested, proving once again that such a delivery method does not affect the physico–chemical properties of the nanosuspensions. 

The release of DCF from nanosuspensions (DCF NS pH 6.0) was also studied before and after the jet injection. As the target site of the injection is the subcutaneous tissue, we prepared a gel (SC gel) mimicking the mechanical properties and the ionic strength of this region, according to a previously published protocol [28]. The bare DCF NS (non-injected) or DCF NS recovered after the NFJI administration across a skin specimen were mixed with the SC gel to simulate the injection site. The mixture was loaded into a dialysis bag, and incubated in a release medium that was sampled over time. In this model, the drug recovered in the release medium represents the drug diffused out of the injection site (subcutaneous tissue) and absorbed into the circulation. A similar system was previously used to model the absorption of monoclonal antibodies upon SC injection, showing good predictive correlations with the in vivo scenario [32]. In agreement with previous results, the release profiles of DCF from jet-injected/non-injected nanosuspensions were not significantly different (Figure 5), showing a cumulative release after 24 h of 52.3 ± 8.0% and 45.7 ± 6.8%.

Collectively, data retrieved from dimensional and morphological analysis, as well as from solubility and in vitro release studies, strongly suggest that DCF NS can be administered by Comfort-in™, a commercial NFJI, without noticeable changes to the properties of the formulation. Since NFJIs are designed and clinically used to administer pharmaceutical solutions, in the second part of this work, we compared the jet injection of DCF NS with that of a DCF sodium solution, analyzing the effect on the skin structure, the delivery efficacy, and the plasmatic profile of DCF upon administration in rats.

SEM was employed to gain information on possible changes in the skin structure upon jet injection. Specifically, skin specimens were jet-injected with a DCF sodium solution (Figure 6A,B) or DCF NS (Figure 6C,D), and imaged immediately after the procedure, without any sample preparation, operating at a variable pressure (VP). SEM analysis in VP mode implies a loss in resolution, but allows for the analysis of biological samples in the hydrated state, returning a picture that is more representative of the natural state of the skin tissue [33]. From this macroscopic observation, it is possible to find that the jet injection punctures the skin, producing an ellipse-like hole with different areas when either the DCF sodium solution (1.13 × 10^5^ µm^2^) or the DCF NS (0.59 × 10^5^ µm^2^) is injected. Such a difference was not investigated further but, as visible in Figure 6A, might be related to the presence of a hair in the trajectory of the jet which could have been split in two, producing a larger hole. What is clearly visible from Figure 6C is the layer of nanocrystals surrounding the injection hole. Despite the liquid jet having sufficient velocity to puncture the skin, a part of the fluid ricochets against the skin and remains on its surface. It is important to take into account such a phenomenon, as this portion of the formulation, since it is not being delivered to the appropriate site beneath the skin, would not trigger any biological effect. 

To determine if the type of formulation influences the amount of drug splashed on the skin, we performed jet injections of a DCF sodium solution or DCF NS, and recovered the liquid on the skin surface by blotting with tissue paper. DCF was then extracted from the tissue and quantified by HPLC, together with the drug extracted from the skin, and the drug recovered from below the skin. Results reported in Table 2 highlight that both formulations (i.e., solution and nanosuspension) have a comparable delivery efficiency to the hypoderm, i.e., higher than 96% of the injected dose. Accordingly, the amount of DCF splashed on the skin when injecting a solution (2.6 ± 1.1%) is not significantly different when injecting a nanosuspension (2.7 ± 1.6%).

Following the comparable delivery efficacy between DCF NS and a solution demonstrated in vitro, the feasibility of administering DCF NS by an NFJI was assessed in vivo. The aim of this experiment was to collect preliminary data on the bioavailability of DCF following the jet injection of a nanosuspension, comparing it with a DCF sodium solution. To this end, male Sprague–Dawley rats were divided in two groups and treated with a single jet injection of either DCF NS or DCF sodium solution (6.65 mg DCF/kg). Blood sampling was performed at fixed time points over the following two hours through an intravenous catheter, and plasmatic DCF was quantified by HPLC.

The administration by Comfort-in™ successfully delivered NS DCF to the subcutaneous region, from which the drug released by the nanocrystals could access the bloodstream for distribution, leading to the concentration versus time curve reported in Figure 7 (grey squares). 

As expected, the jet injection of a DCF sodium solution (blue circles) led to a more rapid decrease in plasma concentration of DCF compared to the nanosuspensions. Indeed, the nanocrystal dissolution represent an additional step which is not present in the case of a solution, where DCF is solubilized in the form of sodium salt, and can be absorbed immediately after the administration. Overall, both DCF NS and DCF sodium solution affected DCF plasma levels in a time-dependent manner. Two-way ANOVA showed a main effect of treatment (F_(1, 7)_ = 9.789; *p* < 0.05), and time (F_(1.942, 13.59)_ = 7.647; *p* < 0.01), but not a significant interaction. Despite the limited number of animals employed and timepoints analyzed, these data provide a preliminary confirmation that NFJIs can be used to administer DCF NS, allowing them to reach therapeutically relevant plasmatic levels and—due to the inherent differences between the dosage forms—a different concentration/time curve than a solution [34].

## 4. Conclusions

The reformulation of well-known drugs with a consolidated clinical use allows to obtain new products with improved biopharmaceutical features, with considerably less hurdles and costs than developing new chemical entities. In this regard, nanotechnology has the potential to revolutionize the drug delivery field, providing tools to increase drug targeting, solubility, stability, and other properties. In this work, we assessed the feasibility of using a clinically approved needle-free jet injector to administer diclofenac nanosuspensions. Our results showed that the nanosuspensions’ features were not altered by the high pressure borne in the injector chamber, nor by the high-speed impact with porcine skin, as the shape, size, solubility, and release profile were maintained upon jet injection. Additionally, we observed that the administration of nanosuspension through a jet injector was as efficient as the administration of a solution, that is, the dosage form most employed with such a device. Finally, the jet injection of nanosuspensions in rats led to therapeutically relevant plasmatic concentrations of diclofenac, confirming the feasibility of the approach in vivo. This work provides the basis for the use of NFJIs with nanosuspensions, a combination that would, on the one hand, broaden the range of applications of NFJI to nanoparticle-based pharmaceuticals, and, on the other, increase patient acceptability for the reduced invasiveness compared to a traditional injection.

## Figures and Tables

**Figure 1 pharmaceutics-14-01085-f001:**
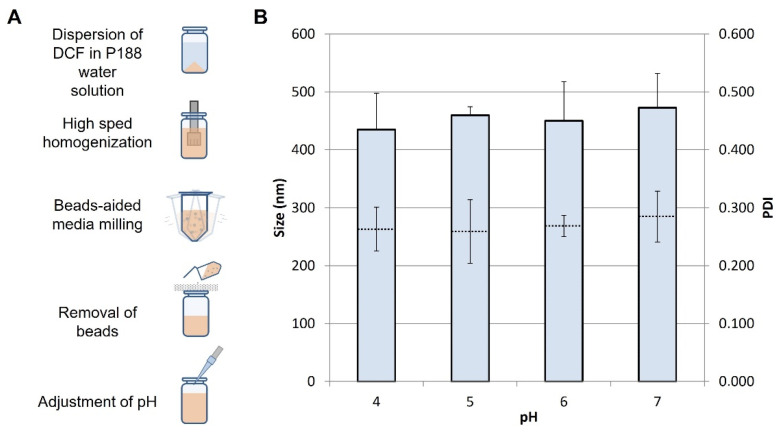
(**A**) Schematic of the preparation method of DCF NS. (**B**) Z average (full bars) and PDI (dotted lines) of DCF NS after pH adjustment at the value reported in the x axis. Values represent the average ± standard deviations of *n* = 3 replicates.

**Figure 2 pharmaceutics-14-01085-f002:**
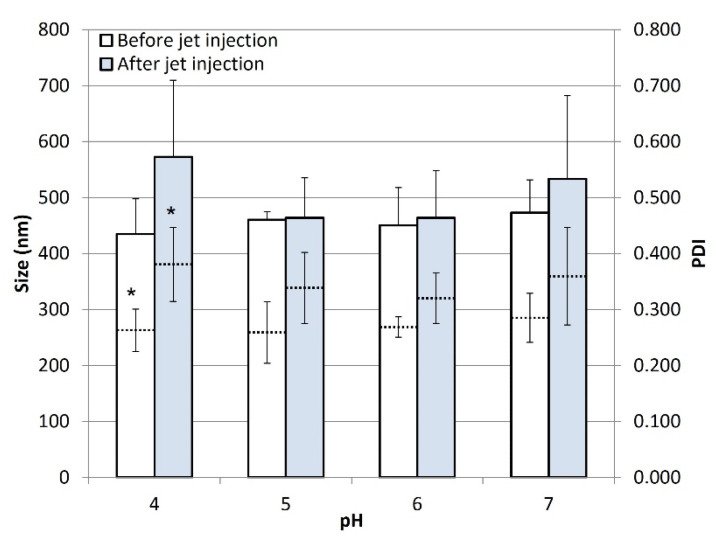
Z average (full bars) and PDI (dotted lines) of DCF NS before and after jet injection across a skin specimen. Values represent the average ± standard deviations of *n* = 3 replicates. Pairs labelled with * indicate a difference with *p* < 0.05. Difference between unlabeled pairs have *p* > 0.05.

**Figure 3 pharmaceutics-14-01085-f003:**
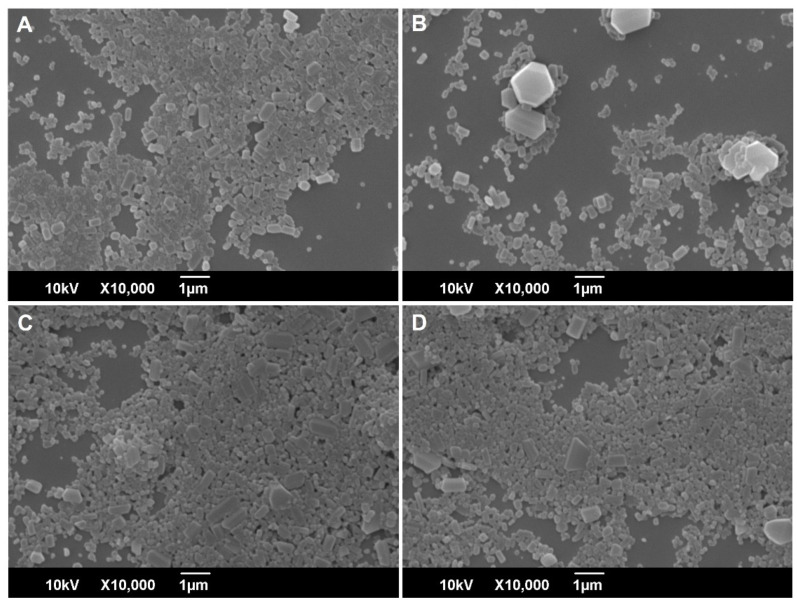
Scanning electron microscopy analysis of DCF NS (pH 6.0) before (**A**,**B**) and after (**C**,**D**) the jet injection across a skin specimen.

**Figure 4 pharmaceutics-14-01085-f004:**
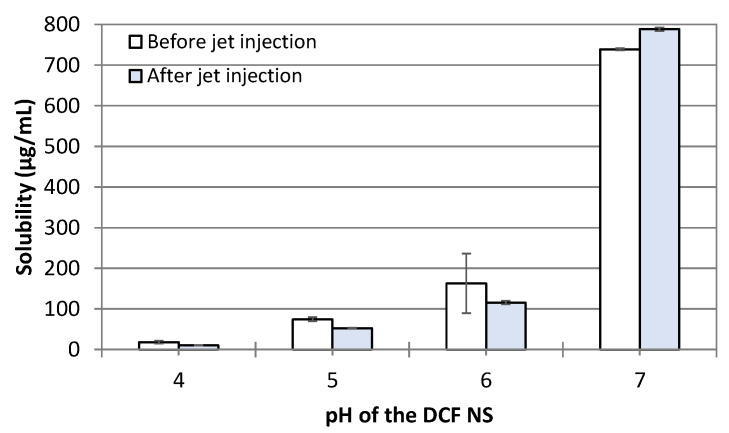
Saturation solubility of DCF NS in deionized water at 37 °C before and after the jet injection across a skin specimen. Bars represent the average ± standard deviations of *n* = 3 independent measurements.

**Figure 5 pharmaceutics-14-01085-f005:**
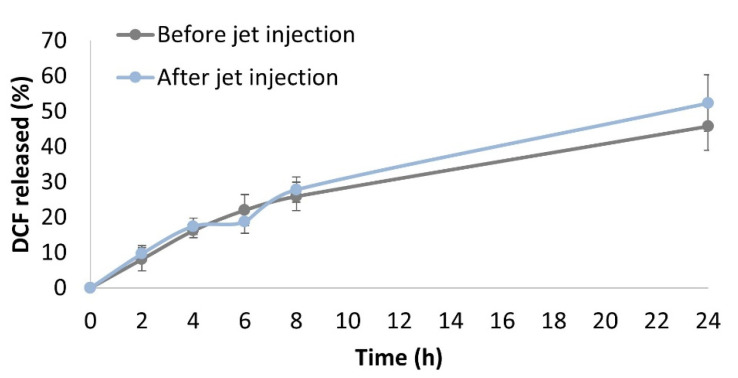
Drug release from the DCF nanocrystals before or after jet injection in a gel matrix simulating the subcutaneous tissue. Points represent the average of *n* = 3 independent experimental determinations.

**Figure 6 pharmaceutics-14-01085-f006:**
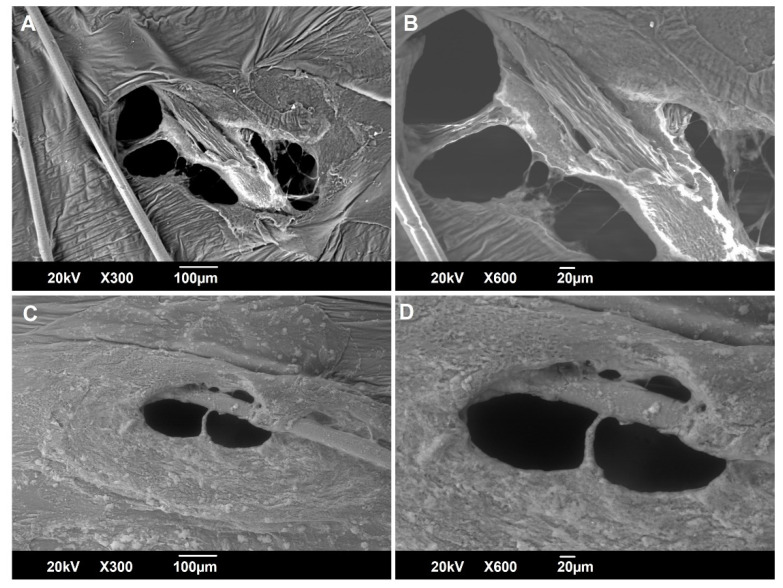
Scanning electron microscopy analysis of newborn pig skin after the jet injection of a DCF sodium solution (**A**,**B**) or DCF NS (pH 6.0) (**C**,**D**).

**Figure 7 pharmaceutics-14-01085-f007:**
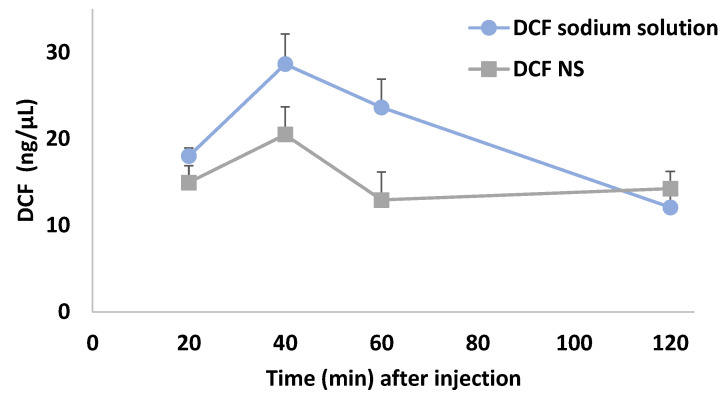
Concentration versus time curves of plasmatic DCF levels. Results are shown as mean ± SEM of the changes in DCF plasma concentrations after administration of DCF sodium solution (blue circles, *n* = 4) or DCF NS (grey squares, *n* = 5). Results analyzed by two-way ANOVA; Bonferroni’s post hoc test.

**Table 1 pharmaceutics-14-01085-t001:** Research works reporting the combined use of nanoparticles with needle-free liquid jet injectors for drug administration purposes and/or to study the impact of formulation and device parameters on the injection results. Abbreviations: ID, intradermal; JI, jet injection.

Nanoparticle	Size	Results	Ref
Cationic a-D-glucan nanoparticles	>70–80 nm	>ID JI of the nanoparticles enhanced the immune response to a protein antigen (pigs, in vivo)	>[19]
>Exosomes	>97 nm, 151 nm, 162 nm	>ID JI of exosomes caused less trauma than a conventional syringe, promoted collagen generation and dermal matrix thickening, and is suitable for cosmetic applications (mice, in vivo)	>[18]
>Liposomes Transfersomes	>55 nm 46 nm	>Liposomes and transfersomes maintained their structural integrity, drug loading, and release properties upon JI (pig skin, in vitro)	>[17]
>Spherical or rod-shaped PLGA particles	>From 0.2 to 25 µm	>Injection volume, standoff distance, and particle size had an effect on the dispersion area and delivery efficiency, while particle shape and concentration did not influence these parameters (human skin, in vitro)	>[15]
>PLGA nanoparticles	>From 45 to 450 nm	>Nozzle diameter, injection pressure, and particle size influenced the penetration depth and the dispersion patterns (mouse skin + acrylamide gel, in vitro)	>[16]
Cationic solid lipid nanoparticles	270 nm	ID JI of pDNA-nanoparticles led to higher antibody titers than the immunization through subcutaneous (20 fold) and topical routes (65 fold) (mice, in vivo)	[14]

**Table 2 pharmaceutics-14-01085-t002:** DCF recovered on the skin surface, within the skin or below the skin (vial) after jet injection of a DCF sodium solution or DCF NS (pH 6.0) across a skin specimen. Values are expressed as the percentage of the DCF dose injected. Results are the average of *n* = 3 independent determinations.

	DCF Sodium Solution	DCF NS
Skin surface	2.6 ± 1.1%	2.7 ± 1.6%
Skin	1.2 ± 0.5%	1.2 ± 0.7%
Vial	96.2 ± 4.0%	96.1 ± 3.3%

## Data Availability

Data are available from the authors upon reasonable request.

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
