# Peer review of "Needle-Free Jet Injectors and Nanosuspensions: Exploring the Potential of an Unexpected Pair"

_pharmaceutics, 2022, doi:10.3390/pharmaceutics14051085_

Round 1
Reviewer 1 Report
This is an interesting paper in which a large number of experimental studies were performed a suitable alternative to conventional syringes by the needle-free jet injectors. Please respond to the following questions and comments.
Major comment:
- Comments:Lines 113. Please indicate the specific value and position of the full thickness skin indicated in the text.
- Comments:Lines 149. Please explain the specific operation method of paper towel and prove that the drug has been completely adsorbed on the paper.
- Comments:Quantification of DCF delivered Please briefly explain whether DCFsodium could stably exist in muscle in nano.
- Comments:Lines 296. Please supplement the drug release curve and set the time cut-off point to the platform period or greater than 95% for evaluation.
- Comments:Figure 7. Although the size and release of DCF NSwere evaluated in vitro, the in vivo results showed that the absorption of DCF NS group was lower than DCF sodium solution group.
- Comments:Various conclusions in the article show a large number of drug residues in subcutaneous tissue. Please briefly explain whether the use of the preparation is safe.
Author Response
We thank the reviewer for the time and effort dedicated to the revision of our manuscript. The questions raised have been addressed and answers are reported in a point-by-point list below. The Reviewer's comments are written in bold, while our responses are given in the ordinary font. Where needed, the manuscript has been amended. All the changes were made using the “Track Changes” function, as indicated by the editor.
Major comment:
1. Comments: Lines 113. Please indicate the specific value and position of the full thickness skin indicated in the text.
We thank the reviewer for the comment. The full thickness skin, obtained from the back part of the animal, was cut in circular specimen with a diameter of 2.5 cm and placed on a wire mesh standing on a glass vial, with the stratum corneum side on the top. These details have been added in the text (lines 115-119).
2. Comments: Lines 149. Please explain the specific operation method of paper towel and prove that the drug has been completely adsorbed on the paper.
To ensure the complete absorption of the drug, each skin specimen was blotted using paper towel measuring 2.5x2.5 cm2. Repeating this procedure with three pieces of paper towel for each skin specimen, the skin surface was completely dried guaranteeing no drug residues on it. The procedure has been clarified in lines 154-158.
3. Comments: Quantification of DCF delivered Please briefly explain whether DCFsodium could stably exist in muscle in nano.
As reported in literature, poorly soluble drugs are often formulated as nanosuspension and administered intramuscularly to obtain a drug nanocrystals depot that slowly dissolve and guarantee a long-acting effect (Ma, Ziwei; Zhang, Hongjuan; Wang, Yanjiao; Tang, Xing (2019). Development and evaluation of intramuscularly administered nano/microcrystal suspension. Expert Opinion on Drug Delivery, 1–15. doi:10.1080/17425247.2019.1588248.). In fact, several products, such as Invega Sustenna® by Janssen, have already been approved by regulatory agencies and are available on the market as injectable drug nanocrystals suspension for intramuscular use.
4. Comments: Lines 296. Please supplement the drug release curve and set the time cut-off point to the platform period or greater than 95% for evaluation.
In principle, we agree with the Reviewer’s comment on the importance to obtain a full in vitro release profile. However -in line with the overall aim of the research – this experiment was designed to assess if the jet injection could have any effect on the release kinetics of DCF from the nanocrystals. As such, not noticing any difference in the release profile of the jet injected and the non-jet injected formulations at the 5 time points analyzed, we decided to stop the experiment after 24 hours.
5. Comments:Figure 7. Although the size and release of DCF NSwere evaluated in vitro, the in vivo results showed that the absorption of DCF NS group was lower than DCF sodium solution group.
We agree with the Reviewer’s observation about the in vivo results. Nevertheless, the duration of the in vivo experiment was set to two hours, a time scale identified as 1) suitable to gain preliminary information on the behavior of nanosuspensions when injected with a needle-free device, and 2) effective in limiting the animal distress due to the blood withdrawals. We are aware that such duration is not sufficient to provide a complete pharmacokinetic profile, that would require additional time points and a higher number of animals, but we consider such study out of the scope of the present work. Coming back to the reviewer observation, we do not exclude that after this time a decrease in DCF sodium plasmatic concentration might occur. On the contrary, DCF NS might guarantee a prolonged constant concentration due to the slow dissolution in the site of injection and the subsequent absorption. These phenomena have already been observed with other nanosuspensions administered through conventional syringes (e.g. reference reported in comment n°3).
6. Comments:Various conclusions in the article show a large number of drug residues in subcutaneous tissue. Please briefly explain whether the use of the preparation is safe.
In our study 400 μl of DCF nanosuspension (10 mg/mL) were used, meaning that a maximum of 4 mg of the drug could be accumulated in the subcutaneous tissue. This is in fact a small amount when compared to what can be achieved by other products containing diclofenac. Indeed, different products having markedly higher concentration of diclofenac (25, 50 or 75 mg/ml) have been approved and are routinely used in the clinic for subcutaneous injection (e.g. Akis® and Dicloin®). Importantly, the use of these products is not associated with adverse effects linked to the amount of drug in the sub-cutaneous tissue. As such, we hope to have convinced the reviewer that the amount of diclofenac delivered by our preparation is safe.
Reviewer 2 Report
Study of applications drug nanosuspensions is important for the clinical praxis and interesting topic for the pharmaceutics. Prepared nanoformulations of diclofenac is stable in the various pH can be by needle-free jet injectors without loss of its nanoform.
Figure 3 is made for the pH 6.0. It wouldn't be pH 7.0/7.3 more suitable?
Figure 7 The study time could be longer.
The results obtained should be compared with other studies, for example in the form of a table.
Author Response
We thank the reviewer for the time and effort dedicated to the revision of our manuscript. The questions raised have been addressed and answers are reported in a point-by-point list below. The Reviewer's comments are written in bold, while our responses are given in the ordinary font. Where needed, the manuscript has been amended. All the changes were made using the “Track Changes” function, as indicated by the editor.
Study of applications drug nanosuspensions is important for the clinical praxis and interesting topic for the pharmaceutics. Prepared nanoformulations of diclofenac is stable in the various pH can be by needle-free jet injectors without loss of its nanoform.
Figure 3 is made for the pH 6.0. It wouldn't be pH 7.0/7.3 more suitable?
We thank the reviewer for the interesting comment. The formulation at pH 6.0 has been chosen since its solubility is lower than the one at pH 7.0, and thus a drug nanocrystals depot that gradually dissolve can be formed, ensuring a slower and constant drug release. Moreover, since the freshly prepared formulation has a pH of 5.8, the adjustment to the pH value of 6.0 requires a lower amount of NaOH, reducing the time of preparation and use of resources (important aspects for a potential scale up).
Figure 7 The study time could be longer.
In principle, we agree with the Reviewer’s observation: a longer experiment could provide additional information on the pharmacokinetics of DCF delivered as a nanosuspension. However, in line with the general scope of the work, this experiment was carried out to gain preliminary data on the feasibility of administering a nanosuspension with the jet injector. As such, a two hours experiment is sufficient to highlight that the nanosuspension is deposited in the subcutaneous tissue and, from this region, molecules of diclofenac can slowly dissolve from the nanocrystals and absorbed in the circulation, with a pharmacokinetic profile that differs from the one of a solution. Moreover, the number of blood sampling/study duration was limited to avoid unnecessary suffering of the animals.
The results obtained should be compared with other studies, for example in the form of a table.
We agree with the reviewer that a table including data from other studies could provide an overview of the state of the art of the combination nanomedicines + jet injectors. Thus, we conducted a thorough research of the literature and drafted a table reporting the previous experimental works where needle free jet injectors were used in combination with nanoparticles. The table was added to the manuscript as Table 1 (and the old Table 1 is re-numbered accordingly), introduced by a comment (lines 70-72). The bibliography was therefore enriched with 5 additional references.
Reviewer 3 Report
This work investigated the possibility of using liquid jet injectors suitable alternative to conventional syringes for the administration of nanosuspensions. Diclofenac stabilized with poloxamer 188 was employed as model for this study. The impact of this novel drug administration technique on the pharmaceutical efficacy and physicochemical properties of the nanosuspension was also analyzed. The in vivo performance of the formulation delivered by the liquid jet injectors was also evaluated using rats.
Minor changes to be made:
- Section 2.2. – This statement “bulk drug (10 mg/ml) was dispersed in an aqueous solution of P188 (5 mg/ml)” is not clear. The authors should please clarify if the bulk drug was fist dissolved in a solvent (specify) at a concentration of 10 mg/mL) prior to dissolution in the aqueous poloxamer. Is the concentration of the aqueous poloxamer 5 mg/mL or the concentration of the drug in the poloxamer solution? This clarification should be included in the manuscript to make comprehension easier.
- Section 2.6. – Please specify the quantity of the clear supernatant that was withdrawn and diluted in methanol for HPLC analysis
- Section 2.8 – Please indicate the pH of the release buffer in the text
- Section 2.9 – “At different time points from the injection”; authors should replace “from” with “after”
- Section 2.9., Reorder this statement – “Adult (weight 275-300 g) male Sprague-Dawley rats (Envigo, Italy)”. Authors may want to consider Adult male Sprague-Dawley rats weighing 275 – 300 g (Envigo, Italy)
Author Response
We thank the reviewer for the time and effort dedicated to the revision of our manuscript. The questions raised have been addressed and answers are reported in a point-by-point list below. The Reviewer’s comments are written in bold, while our responses are given in the ordinary font. Where needed, the manuscript has been amended. All the changes were made using the “Track Changes” function, as indicated by the editor.
Minor changes to be made:
Section 2.2. – This statement “bulk drug (10 mg/ml) was dispersed in an aqueous solution of P188 (5 mg/ml)” is not clear. The authors should please clarify if the bulk drug was fist dissolved in a solvent (specify) at a concentration of 10 mg/mL) prior to dissolution in the aqueous poloxamer. Is the concentration of the aqueous poloxamer 5 mg/mL or the concentration of the drug in the poloxamer solution? This clarification should be included in the manuscript to make comprehension easier.
We thank the Reviewer for the observation about the preparation of the formulation. Firstly, an aqueous solution of P188 was prepared with a concentration of 5 mg/mL P188. The bulk drug was then dispersed in the P188 aqueous solution. Therefore, the obtained suspension had a concentration of 5 mg/mL P188 and 10 mg/mL DCF. For the sake of clarity, paragraph 2.2 has been edited.
Section 2.6. – Please specify the quantity of the clear supernatant that was withdrawn and diluted in methanol for HPLC analysis
For the evaluation of the saturation solubility, 200 μl of clear supernatant have been withdrawn and diluted with 800 ul methanol for HPLC analysis. This information has been added in the main text.
Section 2.8 – Please indicate the pH of the release buffer in the text
After dissolution of the different components of the release buffer, its pH was adjusted to the value of 7.4. Paragraph 2.8 has been modified to include this information.
Section 2.9 – “At different time points from the injection”; authors should replace “from” with “after”
Section 2.9., Reorder this statement – “Adult (weight 275-300 g) male Sprague-Dawley rats (Envigo, Italy)”. Authors may want to consider Adult male Sprague-Dawley rats weighing 275 – 300 g (Envigo, Italy)
We thank the reviewer for the suggestions. The paragraph has been corrected following the indications.
Round 2
Reviewer 1 Report
The questions we proposed have been answered in detail by the authors. Therefore, I think this manuscript is suitable for publication in Pharmaceutics.